# Virological Treatment Monitoring for Chronic Hepatitis B

**DOI:** 10.3390/v14071376

**Published:** 2022-06-24

**Authors:** Elisabetta Loggi, Stefano Gitto, Filippo Gabrielli, Elena Franchi, Hajrie Seferi, Carmela Cursaro, Pietro Andreone

**Affiliations:** 1Hepatology Unit, Department of Medical & Surgical Sciences, University of Bologna, 40126 Bologna, Italy; bettaloggi@gmail.com; 2Department of Experimental and Clinical Medicine, University of Florence, 50134 Florence, Italy; stefano.gitto@unifi.it; 3Postgraduate School of Internal Medicine, University of Modena and Reggio Emilia, 41126 Modena, Italy; judeslash1@gmail.com (F.G.); elenafr6@gmail.com (E.F.); seferihajrie@gmail.com (H.S.); 4Department of Surgical Sciences, University of Bologna, 40126 Bologna, Italy; 5Division of Internal Medicine, Department of Medical and Surgical Sciences, Maternal-Infantile and Adult, University of Modena and Reggio Emilia, 41126 Modena, Italy; carmela.cursaro@virgilio.it; 6Postgraduate School of Allergy and Clinical Immunology, University of Modena and Reggio Emilia, 41126 Modena, Italy; 7Medicina Interna Metabolico-Nutrizionale, Ospedale Civile di Baggiovara, Via Pietro Giardini 1355, 41126 Modena, Italy

**Keywords:** chronic hepatits B, treatment monitoring, serological tests, virological tests

## Abstract

More than 250 million people worldwide are currently infected with hepatitis B, despite the effectiveness of vaccination and other preventive measures. In terms of treatment, new therapeutic approaches are rapidly developing, promising to achieve the elimination of infected cells and the complete cure of infection. The on-treatment monitoring of these innovative antiviral treatments will require the implementation of new virological tools. Therefore, new biomarkers are being evaluated besides the traditional virological and serological assays in order to obtain information on different steps of the viral replication cycle and to monitor response to therapy more accurately. The purpose of this work is to describe both standard and innovative tools for chronic hepatitis B treatment monitoring, and to analyse their potential and feasibility.

## 1. Introduction

Hepatitis B virus (HBV) continues to represent a major global health issue, despite a number of effective measures of control [1]. HBV epidemiology has been dramatically changed by the availability of vaccine prophylaxis, the continued efforts to improve treatment, and the growing awareness of disease [2]. WHO has planned a number of interventions aiming to achieve viral hepatitis eradication by 2030 2014 http://apps.who.int/gb/ebwha/pdf_files/WHA67/A67_R6-en.pdf (accessed on 13 April 2022). Since chronic HBV still affects more than 250 million people worldwide according to recent estimates, however, this goal would seem ambitious [1].

Therapies currently available aim to prevent disease progression, liver cirrhosis, end-stage liver disease and hepatocellular carcinoma (HCC) development; however, the matter of the “cure” of chronic HBV infection is a more complex concept [3]. 

The eradication of the virus in particular remains a challenge because of its peculiar features. In fact, the viral life cycle of HBV is orchestrated by a complex replication apparatus, involving the formation of particularly stable episomal minichromosomes and covalently closed circular DNA (cccDNA) molecules. cccDNA serves as template for transcription and reservoir for replication cycles [4,5]. Furthermore, the viral genome is able to integrate into the host genome, making the infection susceptible of reactivation in certain conditions, even after years of serological and virological suppression. Therefore, while the goal of completely eradicating the virus seems, at the moment, difficult to achieve, the so-called functional cure, represented by the loss of HBsAg (Hepatitis B surface antigen) positivity with anti-HBs (hepatitis B surface antigen antibody) development, is a more realistic end-point and an optimal surrogate. The end-points adequately met by the current therapeutic approaches are long-term suppression of circulating viremia and seroconversion HBeAg/anti-HBe (hepatitis B envelope antigen/antibody) for HBeAg positive patients. Several studies have shown that suppression of HBV-DNA results in biochemical remission and histological improvement, thus decreasing the risk of developing both cirrhosis and HCC [3,6,7,8].

Current approved treatments of chronic HBV can be broadly classified in immunomodulatory agents (standard and Pegylated interferon-α, PegIFN-α), and antiviral agents (nucleoside and nucleotide analogues-NA) [3]. Immunomodulatory agents are administered in a finite course and can lead to the functional “cure”, meaning HBsAg loss. However, this is achieved in a scarce percentage of patients, not exceeding 10%, regardless of HBeAg status [9,10]. Moreover, this treatment choice is limited by low tolerability and high risks of adverse events, with a relatively low acceptance of IFN by physicians and patients as consequence [11]. Nucleoside and nucleotide analogues (NA) inhibit HBV-DNA synthesis via a competitive interaction with the natural substrates of the HBV polymerase, achieving HBV-DNA suppression in the vast majority of compliant patients. Furthermore, currently recommended NAs (tenofovir disoproxil fumarate TDF, Entecavir ETV, and tenofovir alafenamide fumarate TAF) have a high barrier to resistance associated with an excellent safety profile, which has made NAs the mainstay of treatment in most countries. However, the NA’s mechanism of action does not avoid the cccDNA formation and therefore HBV replication rebounds after antiviral therapy is discontinued in most patients, requiring indefinite long-term or even life-long therapy. In this setting, the chance to discontinue NA treatment in carefully selected patients represents a crucial point. There are some encouraging experiences available in literature, but the issue remains controversial [12,13,14,15].

In the last few years, there is been an increase in efforts focusing on new curative strategies, broadly based on the two following approaches: either to target different steps of the viral cycle, or trigger a powerful immune response able to overcome the functional immune exhaustion characterizing the chronic status [16,17].

On the basis of these premises, the monitoring of chronic HBV infection (CHB) antiviral treatment using the appropriate parameters and tools represents a crucial aspect of infection management. The correct integration of those tools, some of which have already been used in clinical practice for years while others have been recently introduced, defines the optimal monitoring strategy.

## 2. Methods

We conducted a non-systematic review article with the following electronic sources: PubMed, MEDLINE, Ovid, Scopus, Google Scholar, and Web of Science. The following search-words were used: “HBsAg”, “qHBsAg”, “qHBcAb”, “HBcrAg”, “HBV-DNA”, “HBV-RNA”, and “cccDNA” alone or in combination with “serology”, “virology”, and “monitoring”. We took into account all the manuscripts reporting human-related data (inclusion criteria) excluding articles without the full text available, not in English language, abstracts, book chapters, and articles published before 1990 (exclusion criteria). 

## 3. Virological Monitoring

### 3.1. HBV-DNA

The detection of circulating viral genome in serum or plasma, HBV-DNA, represents the core of CHB monitoring for current therapies. It is essential at pre-therapeutic assessment in order to allocate the patients in the proper clinical category, and therefore establish eligibility for treatment [3]. In fact, together with biochemical and histological evaluation, a value of HBV DNA > 2000 IU/mL is the threshold for starting antiviral treatment for chronic patients [3,18,19], according to current international guidelines. Patients with cirrhosis should be treated regardless of the viremia amount [3], according to virological response to treatment is defined as persistently undetectable HBV-DNA (by a sensitive polymerase chain reaction PCR assay) for NA-based treatment, or HBV-DNA < 2000 IU/mL after therapy discontinuation in Interferon-α regimens [3]. To be applicable with sufficient reliability in clinical practice, the tests must have a wide range (up to 7 log 10 IU/mL) and a sensitivity of 5–10 IU/mL [3,18,19]. This accuracy is necessary to properly quantify the pre-treatment viral load in patients with high HBV-DNA levels, to use HBV-DNA undetectability as a reliable marker of viral suppression, and to detect early viremia rebounds. Most of the approved HBV-DNA test arrays meet these requirements. These tests have the advantage to use automated or semi-automated platforms with software-assisted analysis, and therefore do not require extra specialized skills from the operator. However, while their level of sensitivity is considered enough for patients management, a point of interest is represented by their performance in terms of limit of detection LoD (lowest amount of target which can be detected but not quantified as an exact value) and limit of quantification LoQ (smallest amount of target which can be measured and quantified with defined precision and accuracy) [20]. The accurate detection and quantification of minimal and residual viremia has, in fact, proven to be clinically relevant in some circumstances, i.e., the identification of best candidates for therapy discontinuation and early identification of patients with a reactivation.

### 3.2. Viral Genotype

HBV is differentiated into many genotypes based on genetic divergence. To date, nine genotypes (A-I) of the HBV genome, and numerous sub genotypes have been defined, clustered in different geographical areas of the world.

Many evidences suggest that viral genotype affects the natural history of the infection, in terms of HBeAg seroconversion rates, severity of liver disease and emergence of mutants [21] According to these evidences, viral genotype A has been associated with a higher risk of developing chronic infection; HBsAg seroclearance is more likely to occur in genotype A and B patients, compared with genotypes C and D patients [22,23]. About HBeAg seroconversion, patients with genotype C achieve the HBeAg seroconversion later than patients with genotype B, and this results in a disease more fast towards fibrosis, cirrhosis and HCC [24]. A close association between genotype C and HCC has been confirmed by more recent observations [25,26].

In terms of antiviral treatment, viral genotype has been an important variable for Interferon alpha based regimens; indeed, Genotype A is associated with significantly higher rates of both HBeAg and HBsAg loss/seroconversion [27,28], and genotype B also, although a lesser extent, identifies potential good responder to IFN treatment. By contrast, genotype seems to have a weaker role in therapy with NA [29]), although anecdotal experience reported HBsAg loss only in TDF treated patients infected with HBV genotypes A and D, while functional cure was not observed in any patients infected with genotype C [30]. 

On the basis of the above evidences, according to AASLD guidelines HBV genotyping can be useful in patients being considered for peg-IFN therapy, but it is not otherwise recommended for routine testing or follow-up of patients with CHB [18].

### 3.3. Viral Resistance Tests

Viral breakthrough is defined as a 1-log 10 (10-fold) increase in serum HBV DNA from nadir during treatment in a patient who had an initial virological response and requires the search for viral variants selected during therapy.

Drug resistance tests are performed with various techniques: the gold standard is the direct sequencing, which allows the detection of all mutations. Other approaches, such as hybridization assays, are easy to perform but can detect only known specific mutations [31].

Since the introduction of NA for CHB treatment, progressive improvements in the efficacy of drugs have profoundly modified the barrier to antiviral resistance [32]. While therapeutic failure and/or viral breakthrough was quite common with the first approved Nas, this occurrence is rare with the current drugs ETV and TDF. In particular, at the present, no typical TDF-resistant mutations have been described [33]. Similarly, if ETV is used as first-line therapy, the rate of resistance is below 1% after 5 years of therapy [34], but is much higher in patients previously treated with “old” drugs. As consequence, the setting of pluritreated patients represents currently the area of use of resistance tests [18].

## 4. Serological Monitoring

### 4.1. HBsAg

Although viremia is the core of treatment monitoring, it is not directly proportional to the number of infected cells [35]. More importantly, the absence of circulating HBV-DNA does not indicate the elimination of cccDNA from hepatocytes, which is responsible for infection perpetuation [36]. Several studies show that 48–52 weeks of therapy induce a negligible, or very slow reduction of cccDNA [37,38]. Therefore, in the last few years research has focused on the detection of potential surrogate markers that are easy to measure, reliable, and able to get information on intrahepatic viral activity with a simple serum measurement. Among them, HBsAg, the historical hallmark of HBV diagnosis, has acquired a renewed attention. As soon as the quantitative test has become available, the quantitative HBsAg (qHBsAg) has gained a key role in the management of CHB. HBsAg is produced from two sources: translation from transcriptionally active cccDNA, and translation from random viral genes transcribed from integrated HBV-DNA sequences in the host genome [39]. HBsAg is part of the envelope of infectious virions, but it also exists in the form of non-infectious sub-viral spheres and filaments, produced in an amount 100-fold to 100,000-fold higher than mature virions [39]. 

Several chemiluminescence based immunoassays for qHBsAg are available on easily-handled semi-automated platforms. Most of them have an analytical sensitivity around 0.05 IU/mL; tests with increased sensitivity (at least one log) are now available, aiming to further improve the accuracy of qHBsAg measurement and the detection of the variants that can be eluded by less sensitive tests [40,41]. 

The interest in qHBsAg arises from the hypothesis that it can be considered a surrogate marker for cccDNA [37], and also from the fact that the suppression of qHBsAg represents the best therapy outcome to date. In addition, large amounts of circulating HBsAg are considered one of the causes of the immune impairment characterizing CHB; the inverse correlation between HBsAg serum levels and anti-HBV T cell response [42] reinforces even more the significance of qHBsAg from an immunological perspective. Regarding its clinical usefulness, qHBsAg has been firstly introduced as additional tool to identify the so-called inactive CHB carriers along with alanaineaminotransferase activity, HBV-DNA and histological activity [43], since combining more criteria provides better positive and negative predictive values for identification of clinical stage. The cut-off qHBsAg value of 1000 IU/mL has been proposed as the best cut-off [43,44]. 

Several studies have investigated the predictive value of qHBsAg kinetics in both IFNα and NA based treatment [45].

Regarding PEG-IFNα therapy, the immunomodulatory effect of PEG-IFNα can induce a robust qHBsAg decline and the role of qHBsAg in optimizing the management seems quite clear, especially in the context of HBeAg positive infection. While qHBsAg baseline values do not seem to have any predictive value, regardless of HBeAg status [46,47], levels of HBsAg below 1500 IU/mL at weeks 12 and 24 were associated with higher rates of response to treatment (defined by recommendations from guidelines) [46,48].

Conversely, qHBsAg levels > 20,000 IU/mL at week 12 or 24, or a decrease < 2 log_10_ may reliably identify patients not responding to treatment, and therefore rules to discontinue therapy have been implemented based on this data [3]. Similar data has been obtained for HBeAg negative CHB: some studies have identified 0.5 log 10 by week 12 and 1 log 10 IU/mL by week 24 as the threshold with the best predictive value for treatment response and/or HBsAg loss [49,50].

A combination of no decrease in HBsAg levels and <2 log 10 IU/mL reduction in serum HBV DNA levels at 12 weeks of PegIFN-α therapy predicts no response to the regimen [47]. This evidence has been implemented as a PegIFN-α regimen discontinuation rule [3].

On the other hand, as far as NA treatment is concerned, the majority of patients achieve undetectable HBV-DNA relatively soon after NA start, and therefore the qHBsAg changes during treatment could represent a measure infection control. However, the decrease of qHBsAg is substantially less pronounced as compared to that potentially achievable with IFN-based regimens. Despite the optimal virological response, exceeding the 90% in practice studies [51] after several years, mean changes in quantitative HBsAg at week 48 from baseline are minimal, and the decline of qHBsAg per year during Entecavir, TDF, and TAF therapy is very slow [52,53,54].

The necessity of lifelong NA treatment has a number of implications in terms of long-term side effects, economic burden, and different reimbursement policies across countries. Thus, the possibility of stopping NA treatment has received increasing attention in the last years, until it was eventually included in guidelines. The selection of best candidates for NA discontinuation relies on predictive values of markers able to predict the virological and clinical relapse and, among them, qHBsAg levels have demonstrated to be effective [13,55,56].

International guidelines uniformly propose HBsAg quantitation for managing peg-IFN therapy, while recommendations for NA therapy require more data [3,18,19].

### 4.2. Serological Monitoring: HBeAg

Data on HBeAg quantification are relatively scarcer, despite the recent introduction of some diagnostic measures from the standardization proposed by WHO in 2013 WHO Expert Committee on Biological Standardization; Collaborative study to establish a World Health Organization international standard for Hepatitis B e antigen. As biomarker during the monitoring of PEGIFN-α therapy, low baseline HBeAg titers were associated with a positive predictive value of HBeAg seroconversion following 48 weeks of therapy. Additionally, a failure of HBeAg titer to decline < 100 PE IU/mL after 24 weeks of therapy was associated with a negative predictive value for HBeAg seroconversion of 96%, a prediction capability stronger than that of serum HBV DNA [57].

In the setting of NA regimens, an early decrease in HBeAg level (at weeks 4 and 12 in patients treated with Entecavir) was predictive of virological response, defined as HBV-DNA undetectability at 48 weeks [58], and lower values of qHBeAg were predictive of HBeAg seroconversion in HBV HIV co-infected patients treated with TDF [59].

### 4.3. Serological Monitoring: HBcrAg

HBcrAg is the newest serological marker yet introduced, as an immunoassay for its measurement has been made recently available. HBcrAg consists of the sum of HBcAg, HBeAg, and p22cr, which is a precore protein from amino acid 28 to at least amino acid 150, and this ensemble is produced from coding the precore/core region [60]. Similarly, to the role assigned to qHBsAg, HBcrAg is currently under investigation to define its capability to reflect intrahepatic virological activity, and therefore monitor the effects of treatment on the infection, when HBV-DNA levels are undetectable and thus no longer informative. The interest in HBcrAg arises from the assumption that its quantification might not be influenced by translation from integrated viral sequences. Hence, HBcrAg quantification may represent a more reliable marker of translational viral activity than qHBsAg.

In particular, HBcrAg has been proven to correlate with intrahepatic cccDNA [61] at an extent superior to that of qHBsAg and HBV DNA [60]. Levels of HBcrAg are different across the different phases of HBV natural history: firstly, HBeAg positive infection displays higher HBcrAg levels compared to HBeAg negative [62]. Additionally, in HBeAg negative patients where two distinct clinical forms coexist (Chronic Hepatitis and Chronic Infection), HBcrAg can help to distinguish between them. Indeed, these two forms can sometimes overlap, and it can be difficult to discriminate, but at the same time necessary to drive the clinical management [3]; recent works have shown that a single measurement of HBcrAg allows an accurate identification of clinical profile of HBeAg negative patients [63,64].

Furthermore, several data from different cohorts have demonstrated that a HBcrAg is an excellent predictor of HCC development [65,66].

In terms of treatment, HBcrAg has been proven to be a useful tool for monitoring treatment and predicting the response in different clinical contexts. A recent work performed on 222 HBeAg-positive patients treated with PEGIFN-α with or without lamivudine reported a more pronounced HBcrAg decline in patients responding to treatment and identified a cut-off value at week 24 for treatment discontinuation. However, this cut-off did not show a better performance when compared to qHBsAg level, which is the marker currently recommend by guidelines. These data were consistent with previous experiences [67], which reported the value of HBcrAg at week 12 as predictor of response, with the identification of cutoffs at week 12 and week 24 with robust negative predictive values. Similarly, a more pronounced decline of HBcrAg was observed in HBeAg negative patients responding to PEGIFNα-2a treatment, albeit with a weaker prediction capability than HBV-DNA or qHBsAg [68]. When the predictive values of qHBsAg and HBcrAg were compared in HBeAg negative patients treated with PEG-IFN, the best performance was obtained with the combined use of both the antigens [69].

In NA therapy setting, several works report a gradual decline in HBcrAg serum levels in both HBeAg positive and HBeAg negative patients [70,71,72], with a wider magnitude in HBeAg positive patients, as expected.

Despite these encouraging results, HBcrAg has not yet been included in the monitoring strategy recommended from guidelines, waiting for more clear evidences about its potential superiority compared to the “traditional” markers [3].

As regards the hot topic of NA discontinuation, a number of key points remain to be clarified, in particular how to optimize the use of qHBsAg and HBcrAg, how to define their different prognostic power, and which variables (i.e., age, sex, ethnicity, viral genotype) could influence the performance of these markers. The possibility to utilize a risk score that combines both HBsAg and HBcrAg has been recently investigated on different studies cohorts, overall showing that lower levels of HBcrAg and HBsAg are associated with favourable outcomes post therapy cessation. Therefore, both markers are helpful in identifying the best candidate to NA suspension [73,74,75,76].

In summary, serological biomarkers are being used progressively more in the classification of CHB phases and in treatment monitoring, in particular in the setting of HBeAg negative infection where the grey zones of classification are frequent, and where the definition of treatment response is more complicated. Even though all these markers are supposed to reflect cccDNA amounts and replicative activity, their relative interchangeability is still matter of debate, as they are part of a complex apparatus that is influenced by several variables. 

Recently, a post hoc analysis of a randomized clinical trial of PEGIFN ± NA aimed to simultaneously evaluate qHBsAg, HBcrAg and HBV-RNA has found qHBsAg to be superior in predicting HBsAg loss in comparison to the other biomarkers [76]. 

Further studies on the impact of different variables on serological markers kinetics are warranted in order to define their optimal fields of application. 

## 5. New Tools for Treatment Monitoring

### 5.1. cccDNA

As stated above, cccDNA is responsible for HBV persistence, and its eradication is the ideal end point of antiviral therapy, because it would mean achieving the sterilizing cure of infection [68]. Based on the cruciality of its role in the maintenance of infection, several efforts are focusing on compounds able to directly degrade cccDNA, or, more indirectly, to interfere with its formation and function.

As consequence, the measurement of cccDNA could provide most useful information on the outcome of infection and its control [5,77], mostly for the innovative therapies which are approaching. Firstly, the cccDNA quantification could provide a safer selection of patients eligible for NA discontinuation based on cccDNA reservoir. Furthermore, it could help define whether anti-HBc-positive and HBsAg-negative patients are protected or susceptible to viral reactivation in conditions where reactivation could occur (liver transplant, immunosuppressive therapies), making the tailoring of antiviral prophylaxis possible.

Data on IFN-α treatment showed cccDNA reduction after 48 weeks of treatment, especially with pegylated formulation [78,79], and a reduction below the limit of detection in about half of study population in patients on long term NA [80]. However, surprisingly, a recent study described virological rebound after NA cessation even in patients with undetectable cccDNA, and further data are needed to evaluate the role of cccDNA in NA stop [81].

Despite its crucial role, quantification of cccDNA remains a challenge due to some important limitations: first, this analysis requires a liver tissue sample which can only be obtained with invasive procedures. Second, the lack of standardized testing methods hampers its implementation in laboratory routine [82]. Some studies have reported the presence of cccDNA in the serum, and its role as monitoring marker [83] but this point is still controversial.

### 5.2. HBV-RNA

Serum HBV-RNA exists in multiple forms. The predominant form is pregenomic RNA (pgRNA), transcribed from cccDNA, which serves as template of both reverse transcription and translation of viral polymerase and core protein. pgRNA is encapsidated into the viral capsid, where it is converted into rcDNA through reverse transcription. Other forms are generated as sub-genomic species [84].

HBV-RNA could represent a biomarker of cccDNA activity. Similarly to other surrogate cccDNA markers, HBV-RNA levels are higher in HBeAg positive infections, and lower in inactive low-viremic infections [85], mirroring HBV-DNA patterns in the different phases of HBV natural history [86].

Of note, the advantage of this marker as monitoring tool is the chance to be measured in serum compartment as pgRNA present in virus-like particles, avoiding the need of invasive procedures to get liver tissue samples. Serum levels of HBV-RNA seem correlated with intrahepatic amounts of pgRNA and cccDNA loads [84,87].

As for PEGIFN-α treatment, baseline HBV-RNA represents a good predictor of virological response, and the kinetics during treatment are a potential element to be included in evaluating the rules for discontinuation of therapy [88]. HBV-RNA testing prior to PEGIFN-based regimens could identify patients with high probability of virological response, and HBV-RNA kinetics may serve to stop treatment in patients infected with HBV genotypes B or C with few chances to achieve response [88]. Furthermore, recent studies demonstrated a progressive reduction of HBV-RNA levels during NA treatment in both HBeAg positive and negative patients [89,90]. Despite the reduction of HBV-RNA during treatment, the majority of HBV-DNA suppressed patients still have detectable HBV-RNA, including a portion of patients with HBsAg loss [86], as consequence of the different effects induced by NA on viral DNA and RNA, which appear to be dissociated: while the DNA production is suppressed, the synthesis of pgRNA is not affected, and it is thus accumulated and released into serum. This is supported by the observation of levels of HBV RNA higher than HBV DNA during treatment [91].

Based on these evidences, similarly to the other cccDNA surrogates, HBV-RNA also has been proposed as predictor of viral relapse and HBV-DNA reactivation after NA discontinuation in both HBeAg-positive and negative patients [92,93]. As is for other surrogates, in HBeAg positive patients HBV pgRNA status predicts the long-term prognoses of patients in terms of HBeAg clearance [94].

HBV-RNA is a really promising candidate for adequate monitoring of intrahepatic viral transcriptional activity, and probably the best tool to measure residual viral activity (which is responsible for virological relapse) in long term HBV-DNA suppressed patients. However, important barriers still limit the implementation of HBV-RNA measurement in routine clinical and laboratory practice, especially technical issues; in fact, although commercial HBV-RNA tests have been recently developed, a standardized, reproducible technique is not available yet.

### 5.3. Anti-HBc Quantification

Anti-HBc antibodies represent one of the traditional elements of diagnosis and patient classification [18]. They are present in infected individuals, both in chronic carrier state and in subjects who already gained HBsAg negativity. Classically considered an hallmark of previous or ongoing exposure to the virus, the role of their quantification has been recently evaluated [95]. Several studies showed that anti-HBc levels vary across the different clinical phases of CHB and are associated with hepatitis activity [96], and has been proposed as non-invasive biomarker for significant liver inflammation in CHB patients [97,98]. Recent data suggest that the correlation between serum anti-HBc and inflammation exists regardless of level of serum ALT, thus representing a noninvasive clinical biomarker also in the difficult setting of patients with normal ALT [18,99]. Furthermore, anti-HBc declined in Peg-IFN and in NA-treated patients (*p* < 0.001), with the lowest levels found in long-term responders who cleared HBsAg subsequently [98]. 

Baseline levels of anti-HBc were predictive of HBeAg seroconversion in both patients treated with PEG-IFNa and Adefovir Dipivoxil [95]. Furthermore, baseline anti-HBc levels were strong predictors of double-negative HBV-DNA and HBV-RNA in patients receiving long-term Entecavir therapy [100]. 

In the setting of NA discontinuation, high levels of anti-HBc at the end of NA treatment together with low level of HBsAg, were associated with a reduced risk of clinical relapse in a median 2.5-year follow-up observation [101].

It would be advisable to further investigate the functional and quantitative relationship between core antigen and antibodies to establish their use in the CHB virological monitoring.

To underline the importance of the immune dynamics of the HBV core protein, several observations have shown that T cell-mediated anti-core response is particularly effective in controlling HBV, since the core protein is the preferential target of the immune response in patients with a more effective infection control [17,102].

## 6. Immune Monitoring

Considering the critical role of the T cell mediated-immune response against the virus, a monitoring tool able to measure and detail the specific immune response would be ideal for an adequate patient characterization.

It is accepted that chronic stage of HBV infection is characterized by a number of T cell dysfunctions, which include up-regulation of co-inhibitory signalling pathways and alterations in metabolic and functional properties [103]. This crucial aspect provides the rationale for the new therapies in development, aiming to gain the control of infection restoring the immune functions, i.e., therapeutic vaccines, tool-like receptor agonists and checkpoint inhibitors [104].

Clinical trials evaluating these new molecules have included the testing of some correlates of immune activity in their design, for example chemokines, cytokines and IFN-stimulated gene (ISG) [105].

The study of fine specificity of virus-specific T cell response, as well as its quantity and strength, is usually realized through cytokine assays. These techniques are based on cytokine (typically IFN-g) measurement after stimulation of T cells with viral sequences, and allow to evaluate a wide protein dimensional spectrum, making this technique also suitable for other larger viruses using relatively few cells. Indeed, this experimental approach has been applied in some immunological sub-studies of more recent immune compounds [106,107].

However, these methodologies are still too sophisticated and a long way from being implemented in a routine lab practice, and remain limited to academic setting and specialized laboratories.

## 7. Concluding Remarks

The landscape of CHB in terms of treatment is rapidly evolving, considering that there are now more than 30 new HBV drugs in the pipeline [16,108]. These new compounds have the challenging objective to become a cure for the chronic stage, an end point scarcely met by any available approach.

In parallel, more sophisticated therapies should be associated to a more sophisticated monitoring approach and, indeed, the tools to allocate patients in definite clinical categories and monitor the efficacy of treatment are also rapidly evolving, aiming at serological markers able to provide more information than the classical serology (qualitative HBsAg) and HBV-DNA. These new biomarkers must reflect the intrahepatic control of viral activity and the measure of immune control during treatment. Some of these new markers are already introduced in the clinical practice, given their advantage of standardization, automation and simple use. Some others are more difficult to be introduced, and more efforts have to be spent in order to address this objective.

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
