# Peer review of "Virological Treatment Monitoring for Chronic Hepatitis B"

_viruses, 2022, doi:10.3390/v14071376_

Round 1
Reviewer 1 Report
This is an important topic and the readers of CHB will benefit from the information in this review.
- In Virological Monitoring section, page-3, line99-101, reference for the tests must have 100 a wide range (up to 7 log10 IU/mL) and a sensitivity of 5-10 IU/mL should be added.
- 2. In Serological monitoring: quantitative HBsAg, HBeAg and HBcrAg section, It may be clearer and easier to read if the titles of each section are more specific.
- 3. In Immune monitoring section, It would be perfect if the content of immune cells and cytokines can be increased.
Author Response
Reviewer #1:
In Virological Monitoring section, page-3, line99-101, reference for the tests must have 100 a wide range (up to 7 log10 IU/mL) and a sensitivity of 5-10 IU/mL should be added.
Reply: Following Reveiwer’s suggestion, the reference has been added.
In Serological monitoring: quantitative HBsAg, HBeAg and HBcrAg section, It may be clearer and easier to read if the titles of each section are more specific.
Reply: Following Reveiwer’s suggestion, the outline of the manuscript, with titles and subtitles of paragraphs, have been modified.
In Immune monitoring section, It would be perfect if the content of immune cells and cytokines can be increased.
Reply: We thank for this suggestion. Otherwise, the immune monitoring is not exactly the focus of the manuscript, based on virological monitoring. We think the immunology will play a crucial role in this setting, and we decided to give a view on the approaching future. However, also for taking into account the suggestions of the second Reviewer, we think is better to maintain this part quite concise.
Reviewer 2 Report
Loggi et al., review the literature for virological treatment monitoring for anti-HBV therapy. The authors have reviewed classical as well as newer markers that are becoming increasingly relevant. However, the authors have ignored several fundamental virological aspects including monitoring of HBV drug resistant mutants and HBV genotypes, which significantly weaken the manuscript. The manuscript may benefit from editing for grammar.
Major comments:
- Identification of drug resistant HBV mutations: The authors completely ignore this critical area relevant to virological monitoring of anti-HBV treatment
- There are notable differences among HBV genotypes in terms of HBV markers. For example qHBsAg levels are higher in HBV genotype A. The baseline virus loads are higher in HBV genotype C compared other genotypes. The authors do not discuss HBV genotypes in the manuscript.
- The author may want to emphasis more on HBV cccDNA activity rather than the HBV cccDNA numbers/quantitation.
- It is important to link which of the markers discussed are part of the AASLD guidelines 2018 for the treatment chronic HBV
- Anti-HBc measurement may be particularly useful in CHB patients with normal ALT levels (https://www.nature.com/articles/s41598-017-03102-3)
- The section on immune monitoring is not directly relevant to the manuscript, as it is titled “Virological treatment monitoring for chronic hepatitis B”. Else, they will also need to discuss ALT/AST levels, imaging etc.
Minor comments:
- Abstract “ The on-treatment monitoring of these new sophisticated approaches will require the implementation of new virological tools” – which approaches? Please rephrase this sentence.
- “ ………loss of HBsAg (Hepatitis B surface antigen) positivity with anti-HBs (hepatitis B surface antigen antibody) development, is a more realistic end-point and an optimal surrogate” I think the authors want to say that this is usually not achieved in most treated individuals, but instead they end up stating the exact opposite. Please rephrase.
- “Several chemioluminescence based immunoassays…….” – spelling “Chemiluminescence”
- “As regards PEG-IFNα, the immunomodulatory effect………” – please rephrase the sentence. It is grammatically incorrect.
- “Conversely, qHBsAg levels > 20000 IU/mL at week 12 or 24, or a decrease < 2 log10 have 100% of chances to identify patients not responding to ……….” should read “Conversely, qHBsAg levels > 20000 IU/mL at week 12 or 24, or a decrease < 2 log10 may identify most patients who not responding to ………”
- “As far as NA therapy is concerned, several works report……..” – grammar
- “New Diagnostic tools” this subheading should read “New tools for monitoring treatment”
- “…….pgRNA is incapsidated 275 into the viral capsid….” It is encapsidated
- “………..may serve to stop treatment in patients infected with HBV genotypes B or C unlike to achieve response [82].” Do the authors mean unlikely? Please rephrase.
Author Response
There are notable differences among HBV genotypes in terms of HBV markers. For example, qHBsAg levels are higher in HBV genotype A. The baseline virus loads are higher in HBV genotype C compared other genotypes. The authors do not discuss HBV genotypes in the manuscript.
Reply: Following Reviewer’s suggestion, the section about genotype has been added (Viral genotype section).
The author may want to emphasis more on HBV cccDNA activity rather than the HBV cccDNA numbers/quantitation.
Reply: This is an important point, however, at the moment, the monitoring with cccDNA, which remains an experimental procedure, is based on kinetics of decrease, and consequently on its quantification. A sentence has been added to deeply clarify this point (cccDNA section).
It is important to link which of the markers discussed are part of the AASLD guidelines 2018 for the treatment chronic HBV.
Reply: Recommendations from AASLD 2018, and also EASL 2017, on the use of markers discussed have been added, where available.
Anti-HBc measurement may be particularly useful in CHB patients with normal ALT levels (https://www.nature.com/articles/s41598-017-03102-3).
Reply: We thank the Reviewer for this relevant suggestion. We added this contribute to the Anti-HBc quantification Section.
The section on immune monitoring is not directly relevant to the manuscript, as it is titled “Virological treatment monitoring for chronic hepatitis B”. Else, they will also need to discuss ALT/AST levels, imaging etc.
Reply: We understand the point of view of the Reviewer. However, in terms of innovative approach, we think the immunology will play a crucial role in this area, and as we would like to give a view on the current situation, but also on the approaching future. So, we would love to maintain this concise section also considering the indication of Reviewer #1 that suggested to expand this section.
Minor comments:
Abstract “The on-treatment monitoring of these new sophisticated approaches will require the implementation of new virological tools” – which approaches? Please rephrase this sentence.
“………loss of HBsAg (Hepatitis B surface antigen) positivity with anti-HBs (hepatitis B surface antigen antibody) development, is a more realistic end-point and an optimal surrogate” I think the authors want to say that this is usually not achieved in most treated individuals, but instead they end up stating the exact opposite. Please rephrase.
Reply: We mean that the elimination of cccDNA is a not realistic end-point, and as alternative the HBsAg loss can be considered the best surrogate end-point.
“Several chemioluminescence based immunoassays…….” – spelling “Chemiluminescence”
“As regards PEG-IFNα, the immunomodulatory effect………” – please rephrase the sentence. It is grammatically incorrect.
“Conversely, qHBsAg levels > 20000 IU/mL at week 12 or 24, or a decrease < 2 log10 have 100% of chances to identify patients not responding to ……….” should read “Conversely, qHBsAg levels > 20000 IU/mL at week 12 or 24, or a decrease < 2 log10 may identify most patients who not responding to ………”
“As far as NA therapy is concerned, several works report……..” – grammar
“New Diagnostic tools” this subheading should read “New tools for monitoring treatment”
“…….pgRNA is incapsidated 275 into the viral capsid….” It is encapsidated
“………..may serve to stop treatment in patients infected with HBV genotypes B or C unlike to achieve response [82].” Do the authors mean unlikely? Please rephrase.
Reply: We apologize for these inaccuracies. We addressed all these points.
Round 2
Reviewer 2 Report
The authors have addressed all the concerns raised. I have no additional comments. The manuscript is suitable for publication in its current form.